# Effect of formulations over a Periodic Capacitated Vehicle Routing Problem with multiple depots, heterogeneous fleet, and hard time-windows

Alejandro Arenas-Vasco[1]*, Juan Carlos Rivera[1], Maria Gulnara Baldoquín[2]

1 Mathematical Modeling Research Group, Escuela de Ciencias Aplicadas e Ingeniería, Universidad EAFIT, Medellín, Antioquia, Colombia, 2 Mathematics and Applications Research Group, Escuela de Ciencias Aplicadas e Ingeniería, Universidad EAFIT, Medellín, Antioquia, Colombia

* aarenas2@eafit.edu.co

**Data Availability Statement:** All relevant data are within the manuscript and the on the GitHub https://github.com/aarenas2/MDHFPCVRP-TW.

## Abstract

This article presents a novel contribution to the Periodic Vehicle Routing Problem (PVRP) by introducing two new problem formulations which differ in the usage of the crucial flow variable. The formulations are tailored to meet the specific demands of the vending machine industry in Medellín, Colombia, and require considering a PVRP with time windows, a heterogeneous fleet, and multiple depots. This scenario, tailored to address real-world complexity and computational challenges, brings to light an exponential surge in integer variables as customer numbers increase. The research presents an analysis of PVRPs that include the four mentioned attributes, compares their similarities, and delves into their nuances. From the analysis it is derived that the variant of the PVRP presented has not been considered previously, taking into account not only these attributes, but also the restrictions involved. Empirical experiments are conducted to examine the intricate interplay between the two proposed formulations, highlighting their impact on the performance of the GUROBI solver. The study provides valuable insights into problem-specific adaptations and algorithmic approaches, emphasizing the significance of the proposed formulations in addressing multifaceted PVRPs. In essence, this research positions the introduction of these two formulations as a pioneering step, offering a new paradigm for approaching the PVRP.

## Introduction

The vehicle routing problem (VRP) has been widely studied in Operations Research field. Nowadays, many papers are written about variants of the most basic VRP (the capacitated VRP or CVRP). Let us define a VRP attribute as an additional layer of complexity considered in the VRP to model different contexts and applications. This paper considers a multi-attribute VRP (MAVRP) that involves multiple periods of time (periodicity), multiple depots, a capacitated heterogeneous fleet, and time windows. We refer to this problem as the multi-depot

**Funding:** The authors want to thank Universidad EAFIT (www.eafit.edu.co) who has financed the research project "Un estudio poliedral y métodos de solución para problemas de enrutamiento de vehículos multi-atributo con múltiples periodos en entornos urbanos" with internal code 953-000020, in which this study has been developed. The authors also want to thank Colfuturo (www.colfuturo.org) who financed partially this project with the grant "Convocatoria 909". All grants were given to A.-V.A.

**Competing interests:** The authors have declared that no competing interests exist.

heterogeneous fleet periodic capacitated VRP with time windows (MDHFPCVRP-TW). According to the conducted literature review, there have been no problems with the four attributes and conditions presented in this paper. Integrating all of these four attributes allows the problem to become more realistic and valuable for some industries.

Periodicity is a precious attribute in the tactical planning of an enterprise. Knowing the routing of the vehicles in a week often is more valuable than routing them day by day because this lets the company adequate driver's shifts, the company's inventory, and customer demands beforehand. It is important to remark that, in Colombia in 2020, transportation represented more than 30% of the logistic costs of enterprises. The previous statistic let us conclude that our research is valuable in the context of Colombian logistics.

As stated by [1], the VRP is $\mathcal{NP}$–hard. As a result, making the problem more complex through adding more characteristics is still $\mathcal{NP}$–hard. For instance, the MDHFPCVRP-TW with only one depot, one period of time, one type of vehicle (homogeneous fleet), and large enough time windows for every client becomes a classical VRP. In the VRP jargon, these characteristics are often called "attributes" [2, 3]. The VRP's main attribute in which this paper is centered is periodicity, as it has been widely studied [4], and makes solving the problem more challenging because a VRP must be solved in each time period, and, generally, a decision in a given time period, affects the environment of the subsequent time periods. For a taxonomy of VRP attributes, the reader is referred to [5].

This paper starts with an overview of the Periodic VRP (PVRP) and its relationship with the considered attributes. Secondly, two formulations for an MDHFPCVRP-TW and the problem description are presented. Then, the performances of both formulations are compared. Lastly, the conclusions of the study are presented.

## Overview of Periodic VRPs

The PVRP was proposed by [6] in a paper addressing garbage collection in New York. The city required garbage trucks to visit some points in the city, but not all of them were visited every day. As a consequence, visits were planned for several time periods, and vehicles perform different routes every period of the time horizon.

But garbage collection is not the only real-life application in which multiple routes must be designed over various time periods. This line of thought applies to large distribution companies, which have to deliver products to large clients on a daily basis but to small clients less frequently. Planning the routes for multiple periods of time means more efficient use of the company's resources, greater customer satisfaction, and a leaner process for warehouses, but it also implies a more complex mathematical model.

In this section, papers related to the PVRP with time windows, multiple depots, and heterogeneous fleet are presented. Then, papers that solve a problem with the four attributes are described and compared to our problem variant. Readers interested in PVRP (VRP over time) can be referred to [7].

### First PVRP papers with time windows, multiple depots, or heterogeneous fleet

The PVRP has been the core of multiple papers in vehicle routing applications due to its adaptability to industrial needs. It is common for large enterprises to plan the routes of their vehicles for multiple periods of time, given that customers present different necessities. Since the PVRP's inception in 1974, it was a matter of time for operation research practitioners to add more attributes to the problem to represent a variety of real-life scenarios. These, to the best of

our knowledge, are the first papers to consider time windows, multiple depots, or a heterogeneous fleet in a PVRP:

It is common in delivery processes for clients to request that their goods be delivered within a specific time during the day. When a routing problem includes this characteristic, it becomes a VRP with time windows (VRP-TW). [8] added time windows to a PVRP and to a multi-depot VRP (separately). The paper presents a formulation for the problem and a tabu search metaheuristic to solve it.

A limitation of the first PVRP is its lack of applicability to giant conglomerates, which can attend their customers using multiple depots. [9] evaluated a real-life case where a large company required serving customers from different distribution centers. The company used to assign clients to specific depots, making their operation easier. Authors proved that allowing some products to be delivered from a more distant depot could benefit the company (mainly when the assigned depot did not have the products available). The authors defined this problem as a multi-product, multi-depot periodic distribution problem.

[10] presented a vehicle routing problem at a supermarket chain in the Netherlands. The supermarket had a heterogeneous fleet of vehicles used to attend their customers over several periods. The authors added heterogeneous fleet and inventory to the PVRP.

## PVRP with time windows, multiple depots, and heterogeneous fleet

The bibliography of vehicle routing problems that have a formulation for PVRP with time windows, multiple depots, and a heterogeneous fleet is scarce. To the best of our knowledge, only four papers comply with this description. Initially, a brief insight into each of them is presented. Then, the different characteristics of the problems introduced in each paper are compared with each other and our own problem variant. The compared characteristics are objective functions, problems' structure (constraints), and solution methods.

**General insights.**   The aforementioned paper of [9] was the first in which the four attributes were combined into a single problem. Besides multiple depots, the industry for which the authors formulated the problem had a distribution process that can also be served from multiple trucks. The clients demanded to be served within a time window and were allowed to choose a preferred pattern of service. For example, in a five-day planning horizon, if a client must be visited once, the user could choose which day it was preferred to receive the company's products. This preference did not represent for the company an obligation, but not complying with it was penalized in the objective function.

More than 10 years later, [11] presented another PVRP with time windows, multiple depots, and a heterogeneous fleet. In this case, it was also motivated by a real-life problem in a brewing company in Mexico that needed to determine the weekly routing for its heterogeneous fleet. It was also possible to lease vehicles when the company's fleet could not meet all the demands. The beer dispatched by the company is sold in returnable bottles that must be collected from the customers when empty. Then, the empty bottles are returned to the depot that served the client. This condition forced every vehicle to be assigned to the same depot in the planning horizon.

[12] presented another PVRP with the studied attributes. The authors named it as the multi-depot multi-period VRP with time windows (MDMPVRP-TW). The authors emphasize the usefulness of considering all the attributes for real-life problems. In the model, a set of vehicles are used to attend the demand of a group of customers in different time periods. Intermediate replenishment is a viable option for vehicles with insufficient inventory to fulfill the demand pending in their route.

[13] presented a novel approach focused on logistic collaboration. The problem was called the two-echelon multi-depot multi-period VRP (2E-MDPVRP). The problem was based on a logistical operation in Chongqing, China, where some logistic companies could share resources for a more efficient process. The first part of the process consists of supplying the inventory to distribution centers from logistic centers. This is done using semi-trailer trucks. Then, from the distribution centers, clients must be served using a heterogeneous fleet of vehicles over multiple periods of time. The clients have a time window for every period they must be attended, and the vehicles must start the routes from the depot in which they finished the route the previous time period.

Three of the four problems are formulated by non-linear models; the exception is [11], who proposed a mixed-integer linear programming model. In the case of [9], the non-linearity is easy to linearize as it refers to the maximum value between a set of possibilities. In the case of the other two articles, linearization is more difficult because they are composed of quadratic functions.

For a simpler presentation of the subsequent tables, the papers are referred to using the convention in Table 1, which also satisfies chronological order.

**Objective functions and solution methods.** The five problems object to our comparison (the four mentioned before and ours) have a lot of things in common. The most prominent feature is that all of them are based on making more efficient an industrial operation. More often than not, this feature is translated into minimizing the operational cost. But, this is the primary objective function only for two of the five analyzed problems.

The solving method is more standardized as, with the only exception being this paper, the four others solved the problem using heuristic methods. The most used was tabu search (TS) with two implementations. Variable neighborhood search (VNS), reactive greedy randomized adaptive search procedure (RGRASP), improved reference point-based non-dominated sorting genetic algorithm-III (IR-NSGA-III), and 3D clustering each were used once.

A summary of the objective functions and solving methods for the five problems compared in this section can be found in Table 2.

**Special characteristics.** So far, the analyzed papers have shared their purpose of making an industrial process more efficient despite the heterogeneity of their objective functions. But, as every problem has its share of uniqueness, the general structure of all of them has essential differences. For a more concise presentation of the text, the differences were grouped into six features: conditions for the time windows, usage of the depots, the structure of the demand of the customers, route conditions, and linearity of the formulation.

Time windows are considered one of the most restrictive constraints, given how they limit the search space. As it can be seen in Table 3, time window constraints are almost standard for all the papers. But, [13] allow multiple time windows in some clients for each time period. Meanwhile, [11] set a time window for the clients depending on the time period. The other three have a constant time window for the customers in every time period. In three out of five,

**Table 1. Paper reference convention.**

| Paper | Convention |
|---|---|
| [9] | $\mathcal{I}$ |
| [11] | $\mathcal{II}$ |
| [12] | $\mathcal{III}$ |
| [13] | $\mathcal{IV}$ |
| This paper | $\mathcal{V}$ |

**Table 2. Objective functions and solving methods for PVRP problems with time windows, multiple depots, and a heterogeneous fleet.**

| Paper | Objective function | Solving method |
|---|---|---|
| $\mathcal{I}$ | Minimize the total outbound distribution costs of all depots over a planning horizon: transportation costs, backordering costs, and penalty costs of assigning patterns other than the desirable ones to customers | Three tabu search (TS) heuristics with different long-term memory applications |
| $\mathcal{II}$ | Minimize the total transportation cost depending on the vehicle and the route | A heuristic based on a reactive greedy randomized adaptive search procedure (RGRASP). |
| $\mathcal{III}$ | Minimize the total distance traversed by the fleet over the planning horizon | A hybrid meta-heuristic combining TS and variable neighborhood search (VNS). |
| $\mathcal{IV}$ | Multi-objective function: Minimize the total logistics operational cost, service waiting time, and number of vehicles | A hybrid heuristic: 3D clustering and improved reference point-based non-dominated sorting genetic algorithm-III (IR-NSGA-III) |
| $\mathcal{V}$ | Minimize the total time elapsed in the operation. | Mixed-integer linear programming |

the depots also have time windows. Lastly, it is essential to remark that only our problem allows the vehicles to stand-by in the client's location before the time window opens.

The consideration of customer demand and service takes on another essential characteristic as it defines the formulation's structure. The most common approach involves determining the demand for the current time period. The only exception is [11], where demand needs to be fulfilled before a specific due date within the planning horizon. Another important aspect is the determination of time periods for each client's visit. Three out of five problems define the time periods for client visits by selecting a feasible visiting pattern for each client. In contrast,

**Table 3. Comparison of the attributes of the papers analyzed.**

| Characteristics / References | $\mathcal{I}$ | $\mathcal{II}$ | $\mathcal{III}$ | $\mathcal{IV}$ | $\mathcal{V}$ |
|---|:---:|:---:|:---:|:---:|:---:|
| Time window per customer fixed for all periods | ✓ | ✓ | ✓ | | ✓ |
| Multiple time windows per period | | | | ✓ | |
| Time window for the depots | | | ✓ | ✓ | ✓ |
| Stand-by times allowed by clients | | | | | ✓ |
| A set of delivery patterns allowable to each customer | ✓ | | ✓ | | ✓ |
| Demand fulfilled before a specific due date in the planning horizon | | ✓ | | | |
| Demand of customer per period | ✓ | | ✓ | ✓ | ✓ |
| Service time | ✓ | | ✓ | | ✓ |
| Interdependence between Depots | ✓ | | | ✓ | ✓ |
| Vehicles are assigned to a depot | ✓ | ✓ | | | |
| Depots can provide service to only a subset of customers | | ✓ | | | |
| Depot capacity | ✓ | ✓ | | | ✓ |
| Intermediate replenishment allowed by a nearby depot if required | | | ✓ | | |
| Multiple products | ✓ | | | | |
| Routes start and end in the same depot | ✓ | ✓ | | ✓ | |
| Routes start in the depot they finished the route in the previous time period | | | ✓ | | |
| Multiple vehicle trips considered per period | ✓ | ✓ | | | |
| One customer per trip | | ✓ | | | |
| Maximum duration of a working period | ✓ | ✓ | | | ✓ |
| Depots have a set of feasible trips to serve a customer | | ✓ | | | |
| Maximum number of vehicles for serving customers per period | | | | ✓ | |
| Maximum time of routes | | | | | ✓ |

[13] establish fixed time periods for serving clients, while [11] request a specific demand amount to be fulfilled before a due date, without fixing the visit's time period. These two papers also share another characteristic: service time is not considered, unlike the other three papers.

The relationship between the trifecta of depots, customers, and vehicles is paramount in the problems' structure. In this characteristic, papers differ a lot from one another. For instance, [12] are the only ones to allow intermediate replenishment in nearby depots if required. Three papers find common ground in limiting the depots' capacity. Of those three, [9, 11] also assign a depot to each vehicle in the planning horizon, but the latter also limits the depots to attend only a subset of customers. The other paper that limits the capacity of the depot is ours. Lastly, [9, 13] allow depots to work interdependently, meaning two or more depots can serve the same customer in a time period.

The route's structure is the most complex feature of a routing problem as it has to suit every real-life application's particular situation. [9, 11], and our problem coincide in two route characteristics: the vehicles start and finish each day in the same depot, and every time period has a maximum duration time span (our problem also has a maximum route duration). The former two problems also allow each vehicle to perform multiple trips. The formulation of [11] restricts routes to attend only one customer per trip, and the graph is not complete, so routes can only use certain arcs that connect customers and depots. In the case of the formulation proposed by [12], routes must start from the depot in which they ended their route in the previous time period. Meanwhile, [13] restrict the number of vehicles allowed to attend customers at every time period.

Finally, the problem presented in [9] considers multiple types of products being transported.

As it can be seen in this section, although there exist four studies about vehicle routing with the four studied attributes, all of them differ from each other in some characteristics. In addition, the presented study differs considerably from the other four problems.

## Proposed problem variant

The proposed problem represents a real-life case study in the vending machine service industry in Medellín, Colombia. A company must supply its vending machines efficiently during a time horizon. Every vehicle starts a route at a depot where products are loaded, with which the vending machines are filled. On every visit, the cash in the vending machine must be retired so the vehicle must return to the same depot at the end of the route to deposit the collected cash. The vehicles are outsourced and do not stay parked in the depots at night. The demand of the problem is considered in terms of the number of vending machines in each client. The depots are limited to serving a certain amount of vending machines daily. There are several service patterns for all the clients, but the pattern assignation must comply with the total number of visits required.

Additionally, each client must be attended in a time window that does not change in the planning horizon. Clients also have an invariant service time for every visit in the planning horizon. Vehicles are allowed to arrive before the opening of the time window of every client, but the service cannot start until its opening. These stand-by times make the total time of the routes longer. Each client has a maximum allowed stand-by time, the same for all its visits in the planning horizon. For some of the clients, their stand-by time may be zero, that is, the stand-by time is not allowed.

The fleet of vehicles to meet the demand of the clients is heterogeneous. The difference between types of vehicles is the maximum number of vending machines each can service. In

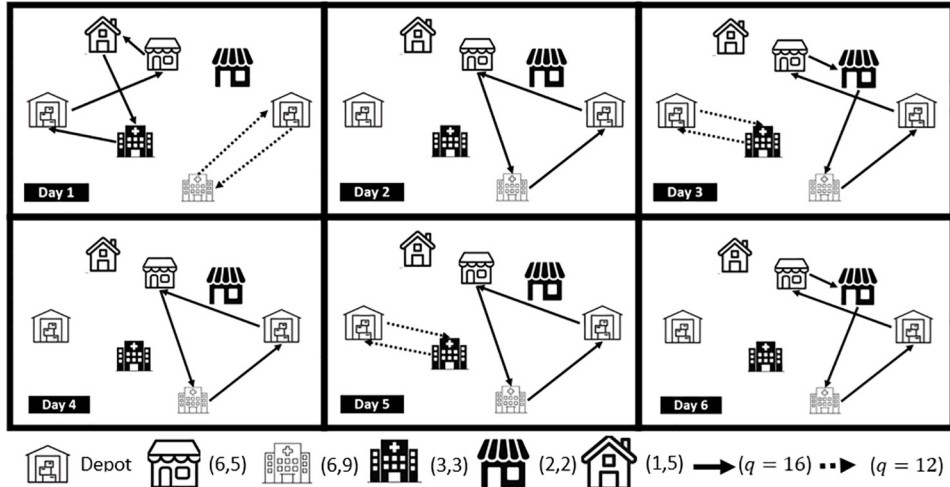

**Fig 1. Illustration of the problem.**

Fig 1 an illustrative example of the routes planned over a six-day period is included. Two parameters of each client are presented as *(Number of visits, demand)*, and the capacity ($q$) of a vehicle traversing an arc is represented with the continuity of the arrow that represents the used arc.

A solution must represent the pattern of visits of every client and the daily: routes of every vehicle, the arrival time of vehicles to the different clients, stand-by times in every client for each visit, and the depot in which every vehicle starts and finishes each route.

The problem is a MDHFPCVRP-TW which objective is to minimize the total time of all the routes (from the departure until the arrival at a depot) in a 6-day span.

For the MDHFPCVRP-TW described above, two formulations are presented in this paper. Both formulations are equivalent being the main difference the number of indexes of the classical flow variable ($x$). The first formulation has four indexes: two for nodes, one for vehicles, and one for the time period. It determines if a vehicle uses an arc thus, $x_{ij}^{kh}$ determines if vehicle $k$ traverses arc $(i, j)$ on day $h$. We refer to this formulation as the 4-index-formulation (4IF).

The second formulation only requires three indexes for the flow variable ($x$): two for nodes and one for the time period. Similarly, $x_{ij}^{h}$ determines if arc $(i, j)$ is traversed in time period $h$ while a combination of new variables and constraints is needed to determine which vehicle performs each route. We refer to this formulation as the 3-index-formulation (3IF).

Now, the sets and the parameters are presented. Both are common to the two formulations. At the end of this section, both formulations are included.

The formulations use the following sets:

- $C$ is the set of clients, where $|C| = n$.

- $CP_f$ is the set of clients with frequency $f$, where $C = \cup_{f \in F} CP_f$ and $\cap_{f \in F} CP_f = \emptyset$

- $F$ is the set of possible frequencies of visit.

- $P_f$ is the set of possible patterns of visit of frequency $f$.

- $D$ is the set of depots.

- $N$ is the set of all the nodes and is conformed by clients and depots ($N = C \cup D$).

- $K$ is the set of vehicles.

- $H$ is the set of days. As referenced before, the scheduling span is of 6 days.

  The formulations also use the following parameters:

- $q_k$ is the capacity of vehicle $k$ which indicates the number of vending machines that can attend daily.

- $dem_c$ is the number of vending machines of client $c$ that must be supplied in each visit.

- $s_c$ is the time to serve client $c$.

- $[a_i, b_i]$ is the time window in which client $i$ must be attended (the upper bound for arriving at a depot is the maximum time duration allowed for a route).

- $l_c$ is the maximum stand-by time allowed in client $c$.

- $t_{ij}$ is the time to traverse arc $(i, j)$ with any vehicle.

- $R_d$ is the number of vending machines that depot $d$ can attend daily.

- $vis_f$ is the number of visits that must receive a client with frequency $f$ in the planning horizon.

- $A_p^h$ is an element of matrix $A$, which relates patterns with the days a client is visited. It takes the value 1 if pattern $p$ forces clients to be visited on day $h$ and 0 otherwise.

- A largely enough value (**M**) is used. According to the types of restrictions where M is used, it could be taken as the upper bound of time window of the depots.

**Mathematical formulation of the 4IF**

The complete formulation of the 4IF is presented. Let us start with the variables:

- $x_{ij}^{kh} \in \{0, 1\}$: binary variable which takes the value 1 if the arc $(i, j)$ is used by vehicle $k$ in the day $h$, and 0 otherwise.

- $u_{cp} \in \{0, 1\}$: binary variable which takes the value 1 if pattern $p$ is assigned to client $c$.

- $T_{ij}^h \geq 0$: time at which a vehicle arrives to node $j$ coming from node $i$ on day $h$.

- $f_{ij}^{kh} \geq 0$: load of vehicle $k$ while traversing arc $(i, j)$ on day $h$.

- $y_c^h \geq 0$: stand-by time of the vehicle visiting client $c$ on day $h$.

  Model (1) to (22) presents the 4IF.

$$\min \quad Z = \sum_{h \in H} \sum_{i \in N} \sum_{j \in N} \sum_{k \in K} ((t_{ij} + s_j) \cdot x_{ij}^{kh}) + \sum_{c \in C} \sum_{h \in H} y_c^h \tag{1}$$

$$\text{s.t.} \quad \sum_{h \in H} \sum_{\substack{i \in N \\ i \neq c}} \sum_{k \in K} x_{ic}^{kh} = vis_f, \quad \forall f \in F, \ c \in CP_f \tag{2}$$

$$\sum_{\substack{i \in N \\ i \neq c}} \sum_{k \in K} x_{ic}^{kh} = \sum_{p \in P_f} A_p^h \cdot u_{cp}, \quad \forall f \in F, \ c \in CP_f, \ h \in H \tag{3}$$

$$\sum_{p \in P_f} u_{cp} = 1, \qquad \forall f \in F, \ c \in CP_f \tag{4}$$

$$\sum_{c \in C} \sum_{d \in D} x_{dc}^{kh} \leq 1, \qquad \forall \ h \in H, \ k \in K \tag{5}$$

$$\sum_{\substack{i \in N \\ i \neq c}} x_{ic}^{kh} - \sum_{\substack{j \in N \\ j \neq c}} x_{cj}^{kh} = 0, \quad \forall \ c \in C, \ h \in H, \ k \in K \tag{6}$$

$$\sum_{c \in C} \sum_{k \in K} f_{dc}^{kh} \leq R_d, \qquad \forall \ d \in D, \ h \in H \tag{7}$$

$$f_{ij}^{kh} \leq q_k \cdot x_{ij}^{kh}, \qquad \forall \ h \in H, \ d \in D, \ j \in C, \ k \in K \tag{8}$$

$$\sum_{\substack{i \in N \\ i \neq c}} T_{ic}^h + y_c^h \geq a_c \cdot \sum_{\substack{i \in N \\ i \neq c}} \sum_{k \in K} x_{ic}^{kh}, \quad \forall \ c \in C, \ h \in H \tag{9}$$

$$\sum_{\substack{i \in N \\ i \neq c}} T_{ic}^h + y_c^h \leq (b_c - s_c) \cdot \sum_{\substack{i \in N \\ i \neq c}} \sum_{k \in K} x_{ic}^{kh}, \quad \forall \ c \in C, \ h \in H \tag{10}$$

$$y_c^h \leq l_c, \qquad \forall \ c \in C, \ h \in H \tag{11}$$

$$\sum_{\substack{i \in N \\ i \neq c}} T_{ic}^h + y_c^h + s_c + t_{cj} - M \cdot \left(1 - \left(\sum_{k \in K} x_{cj}^{kh}\right)\right) \leq T_{cj}^h, \quad \forall \ c \in C, \ h \in H, \ j \in N, c \neq j \tag{12}$$

$$\sum_{\substack{i \in N \\ i \neq c}} T_{ic}^h + y_c^h + s_c + t_{cj} + M \cdot \left(1 - \left(\sum_{k \in K} x_{cj}^{kh}\right)\right) \geq T_{cj}^h, \quad \forall \ c \in C, \ h \in H, \ j \in N, c \neq j \tag{13}$$

$$T_{dc}^h \geq t_{dc} \cdot \sum_{k \in K} x_{dc}^{kh}, \qquad \forall \ c \in C, \ d \in D, \ h \in H \tag{14}$$

$$T_{dc}^h \leq (b_c - s_c) \cdot \sum_{k \in K} x_{dc}^{kh}, \qquad \forall \ c \in C, \ d \in D, \ h \in H \tag{15}$$

$$\sum_{h\in H}\sum_{c\in C}\sum_{d\in D} f_{cd}^{kh} = 0, \qquad \forall\; k \in K \tag{16}$$

$$\sum_{\substack{i\in N \\ i\neq c}}(f_{ic}^{kh} - f_{ci}^{kh}) = dem_c \cdot \sum_{\substack{j\in N \\ j\neq c}} x_{cj}^{kh}, \quad \forall\; c \in C,\; h \in H,\; k \in K \tag{17}$$

$$\sum_{c\in C} x_{cd}^{kh} = \sum_{c\in C} x_{dc}^{kh}, \qquad \forall\; d \in D,\; h \in H,\; k \in K \tag{18}$$

$$x_{ij}^{kh} \in \{0,1\}, \qquad \forall\; h \in H, i \in N, j \in N, k \in K, i \neq j \tag{19}$$

$$u_{cp} \in \{0,1\}, \qquad \forall f \in F,\; c \in CP_f,\; p \in P_f \tag{20}$$

$$0 \leq T_{ic}^h \leq (b_c - s_c), \qquad \forall\; c \in C,\; h \in H,\; i \in N, i \neq c \tag{21}$$

$$y_i^h \geq 0, \qquad \forall\; h \in H,\; i \in N \tag{22}$$

The objective function (1) minimizes the total time of all routes, specifically, it considers traveling times, service times, and stand-by times. As the operating cost of the enterprise is principally based on time spent on a route, all times are given the same importance.

Constraints (2) to (4) define that every client is visited the number of times that their frequency dictates with a feasible pattern: (2) define the number of visits of every client, (3) force one route to visit a client the day the chosen pattern indicates it must be visited, (4) guarantee that each client has an assigned pattern of visit. Constraints (5) and (6) ensure that flow continues through the network: (5) define that each vehicle, on each day that a route starts, must start in a depot and (6) ensure that every client on any day is left after that visit. Constraints (7) guarantee that the total demand attended from a depot on any day does not exceed its capacity. Constraints (8) limit the load of the vehicles to the maximum capacity of each one on any day. Constraints (9) to (11) condition the arrival time of the vehicles in the time window of every client, and the stand-by time of the vehicles: (9) make the model consider the stand-by time of every vehicle and the service time of the client upon arrival, (10) make the vehicles arrive before they cannot service the client without violating the time window, and (11) force the vehicles on any day to comply with the maximum stand-by time allowed in every client. Constraints (12) to (15) actualize the time of arrival to every node. (12) and (13) constraints add to the arrival time of the previous client its stand-by time, service time, and transportation time to the actual node when stand-by times are allowed. (14) guarantee that the arrival time to the first client from a depot considers the time required to travel from the depot to this first client. (15) force the time of arrival to a client from a depot to be zero if the client is not visited directly from that depot. Vehicles are not forced to leave the depot on time zero so, the vehicle's departure time is the time of arrival to the first client minus the time required to transit the arc between that client and the origin depot. Constraints (16) and (17) are the subtour elimination constraints and actualize the load of demand to be attended in every visit. (16) guarantee that the vehicles arrive at every depot daily without any load and (17) subtract the demand of a served client from the load of the following used arc. Constraints (18) force the routes on any day to finish at the depot they started.

Constraints ([19]) to ([22]) indicate the domain of the decision variables.

## Mathematical formulation for the 3IF

The 3IF is equivalent to the 4IF, but instead of defining decision variables in terms of if arc $(i, j)$ is traversed by vehicle $k$ on the day $h$, it only defines if arc $(i, j)$ is traversed on the day $h$. A new set of variables must be defined to link the usage of a vehicle with the usage of arcs. This formulation reduces the number of variables but requires more linking constraints.

Some variables are exactly the same as in the 4IF: $u_{cp}$, $T_{ij}^h$, and $y_c^h$. Let us define the variables that are particular for the 3IF:

- $x_{ij}^h \in \{0, 1\}$: binary variable which takes the value 1 if the arc $(i, j)$ is used in the day $h$, and 0 otherwise.

- $w_i^{kh} \in \{0, 1\}$: binary variable which takes the value 1 if node $i$ is visited on day $h$ by vehicle $k$ and 0 otherwise.

- $r_{dc}^h \in \{0, 1\}$: binary variable which takes the value 1 if a vehicle serving customer $c$ on the day $h$ starts at depot $d$ and 0 otherwise.

- $f_c^h$: demand to be supplied to customer $c$ on day $h$.

Model ([23]) to ([55]) presents the 3IF.

$$\min \quad Z = \sum_{h \in H} \sum_{i \in N} \sum_{j \in N} ((t_{ij} + s_j) \cdot x_{ij}^h) + \sum_{c \in C} \sum_{h \in H} y_c^h \tag{23}$$

$$\text{s.t.} \quad \sum_{h \in H} \sum_{\substack{i \in N \\ i \neq c}} x_{ic}^h = vis_f, \quad \forall f \in F, \ c \in CP_f \tag{24}$$

$$\sum_{\substack{i \in N \\ i \neq c}} x_{ic}^h = \sum_{p \in P_f} A_p^h \cdot u_{cp}, \quad \forall f \in F, \ c \in CP_f, \ h \in H \tag{25}$$

$$\sum_{p \in P_f} u_{cp} = 1, \quad \forall f \in F, \ c \in CP_f \tag{26}$$

$$\sum_{\substack{i \in N \\ i \neq c}} x_{ic}^h - \sum_{\substack{j \in N \\ j \neq c}} x_{cj}^h = 0, \quad \forall c \in C, \ h \in H \tag{27}$$

$$w_d^{kh} + w_c^{kh} - 1 \leq r_{dc}^h, \quad \forall c \in C, \ d \in D, \ h \in H, \ k \in K \tag{28}$$

$$\sum_{d \in D} r_{dc}^h \leq 1, \quad \forall c \in C, \ h \in H \tag{29}$$

$$\sum_{c \in C} dem_c \cdot r_{dc}^h \leq R_d, \quad \forall d \in D, \ h \in H \tag{30}$$

$$\sum_{c \in C} dem_c \cdot w_c^{kh} \leq q_k, \qquad \forall \ h \in H, \ k \in K \tag{31}$$

$$\sum_{\substack{i \in N \\ i \neq c}} T_{ic}^h + y_c^h \geq a_c \cdot \sum_{\substack{i \in N \\ i \neq c}} x_{ic}^h, \quad \forall \ c \in C, \ h \in H \tag{32}$$

$$\sum_{\substack{i \in N \\ i \neq c}} T_{ic}^h + y_c^h \leq (b_c - s_c) \cdot \sum_{\substack{i \in N \\ i \neq c}} x_{ic}^h, \quad \forall \ c \in C, \ h \in H \tag{33}$$

$$y_c^h \leq l_c, \qquad \forall \ c \in C, \ h \in H \tag{34}$$

$$\sum_{\substack{i \in N \\ i \neq c}} T_{ic}^h + y_c^h + s_c + t_{cj} - M \cdot (1 - x_{cj}^h) \leq T_{cj}^h, \quad \forall \ c \in C, \ h \in H, \ j \in N, c \neq j \tag{35}$$

$$\sum_{\substack{i \in N \\ i \neq c}} T_{ic}^h + y_c^h + s_c + t_{cj} + M \cdot (1 - x_{cj}^h) \geq T_{cj}^h, \quad \forall \ c \in C, \ h \in H, \ j \in N, c \neq j \tag{36}$$

$$T_{dc}^h \geq t_{dc} \cdot x_{dc}^h, \qquad \forall \ c \in C, \ d \in D, \ h \in H \tag{37}$$

$$T_{dc}^h \leq (b_c - s_c) \cdot x_{dc}^h, \qquad \forall \ c \in C, \ d \in D, \ h \in H \tag{38}$$

$$f_c^h = dem_c \cdot \sum_{k \in K} w_c^{kh}, \qquad \forall \ c \in C, \ h \in H \tag{39}$$

$$\sum_{c \in C} x_{dc}^h = \sum_{k \in K} w_d^{kh}, \qquad \forall \ d \in D, \ h \in H \tag{40}$$

$$\sum_{c \in C} x_{cd}^h = \sum_{k \in K} w_d^{kh}, \qquad \forall \ d \in D, \ h \in H \tag{41}$$

$$\sum_{\substack{i \in N \\ i \neq c}} x_{ic}^h = \sum_{k \in K} w_c^{kh}, \quad \forall \ c \in C, \ h \in H \tag{42}$$

$$\sum_{k \in K} w_c^{kh} \leq 1, \qquad \forall \ c \in C, \ h \in H \tag{43}$$

$$\sum_{d \in D} w_d^{kh} \leq 1, \qquad \forall \ h \in H, \ k \in K \tag{44}$$

$$\sum_{c \in C} w_c^{kh} \leq n \sum_{d \in D} w_d^{kh}, \qquad \forall\ h \in H,\ k \in K \tag{45}$$

$$x_{ij}^h \leq 1 - w_i^{kh} + w_j^{kh}, \qquad \forall\ i,j \in C, i \neq j,\ k \in K,\ h \in H \tag{46}$$

$$x_{dc}^h \leq r_{dc}^h, \qquad \forall\ c \in C,\ d \in D,\ h \in H \tag{47}$$

$$x_{cd}^h \leq r_{dc}^h, \qquad \forall\ c \in C,\ d \in D,\ h \in H \tag{48}$$

$$x_{ij}^h \in \{0,1\}, \qquad \forall\ h \in H,\ i,j \in N, i \neq j \tag{49}$$

$$u_{cp} \in \{0,1\}, \qquad \forall f \in F,\ c \in CP_f,\ p \in P_f \tag{50}$$

$$r_{dc}^h \in \{0,1\}, \qquad \forall\ c \in C,\ d \in D,\ h \in H \tag{51}$$

$$w_i^{kh} \in \{0,1\}, \qquad \forall\ i \in N,\ h \in H,\ k \in K \tag{52}$$

$$f_c^h \geq 0, \qquad \forall\ c \in C,\ h \in H \tag{53}$$

$$0 \leq T_{ic}^h \leq (b_c - s_c), \qquad \forall\ c \in C,\ h \in H,\ i \in N, i \neq c \tag{54}$$

$$y_i^h \geq 0, \qquad \forall\ h \in H,\ i \in C \tag{55}$$

The objective function (23) is equivalent to (1). Constraints (24) to (26) are equal to constraints (2) to (4). Constraints (27) is equal to constraints (6). Constraints (28) guarantee that a depot can serve a client only if both are associated with the same vehicle each day. Constraints (29) guarantee that each client can be assigned to a maximum of one depot daily. Constraints (30) guarantee that the total demand attended from a depot on any day does not exceed its capacity. Constraints (31) limit the load of the vehicles on any day to the maximum capacity of each one. Constraints (32) to (38) are equal to constraints (9) to (15). Constraints (39) guarantee that a client's demand is satisfied if it is visited that day.

Constraints (40) and (41) condition the start and ending of each route each day in a depot. Constraints (42) force a client to be assigned to a route each day it is visited. Constraints (43) guarantee that any client on any day is visited by, at most, one vehicle. Constraints (44) force the vehicles to be attended by only one depot daily. Constraints (45) let a vehicle attend a client a day only if assigned to a depot that day. Constraints (46) guarantee that there is not a change of vehicle during a route a day (In the S1 Appendix, we demonstrate this constraint is sufficient to guarantee the consistency of the route). Constraints (47) guarantee that a vehicle can depart to a client from a depot on any day only if the client is served from that depot that day. Constraints (48) guarantee that a vehicle can return to a depot from a client on any day only if the client is served from that depot that day.

Constraints (49) to (55) indicate the domain of the decision variables.

Let us compare the number of integer variables in both models. Table 4 compares both formulations' integer variables given some conditions. It can be seen that the variables in both formulations present an exponential growth as $n$ and $|K|$ are augmented (the other parameters

**Table 4. Integer variables in both formulations.**

| n | \|D\| | \|K\| | \|H\| | \|P\| | 4IF integer variables | 3IF integer variables |
|---|---|---|---|---|---|---|
| 14 | 2 | 2 | 6 | 12 | 3 240 | 2 040 |
| 20 | 2 | 2 | 6 | 12 | 6 048 | 3 624 |
| 24 | 2 | 2 | 6 | 12 | 8 400 | 4 920 |
| 28 | 2 | 5 | 6 | 12 | 27 336 | 6 912 |
| 32 | 2 | 5 | 6 | 12 | 35 064 | 8 664 |
| 36 | 2 | 5 | 6 | 12 | 43 752 | 10 608 |
| 40 | 2 | 7 | 6 | 12 | 74 568 | 13 224 |
| 44 | 2 | 7 | 6 | 12 | 89 400 | 15 600 |
| 48 | 2 | 7 | 6 | 12 | 105 576 | 18 168 |
| 52 | 2 | 9 | 6 | 12 | 158 088 | 21 552 |
| 56 | 2 | 9 | 6 | 12 | 182 328 | 24 552 |
| 60 | 2 | 9 | 6 | 12 | 208 296 | 27 744 |

were not changed), but the number of integer variables on the 4IF has the steepest slope. This can be seen clearer in Fig 2.

## Performance comparison between formulations

Having two equivalent formulations does not mean that both perform equally. This section presents the results of multiple computational experiments solving various data sets of the

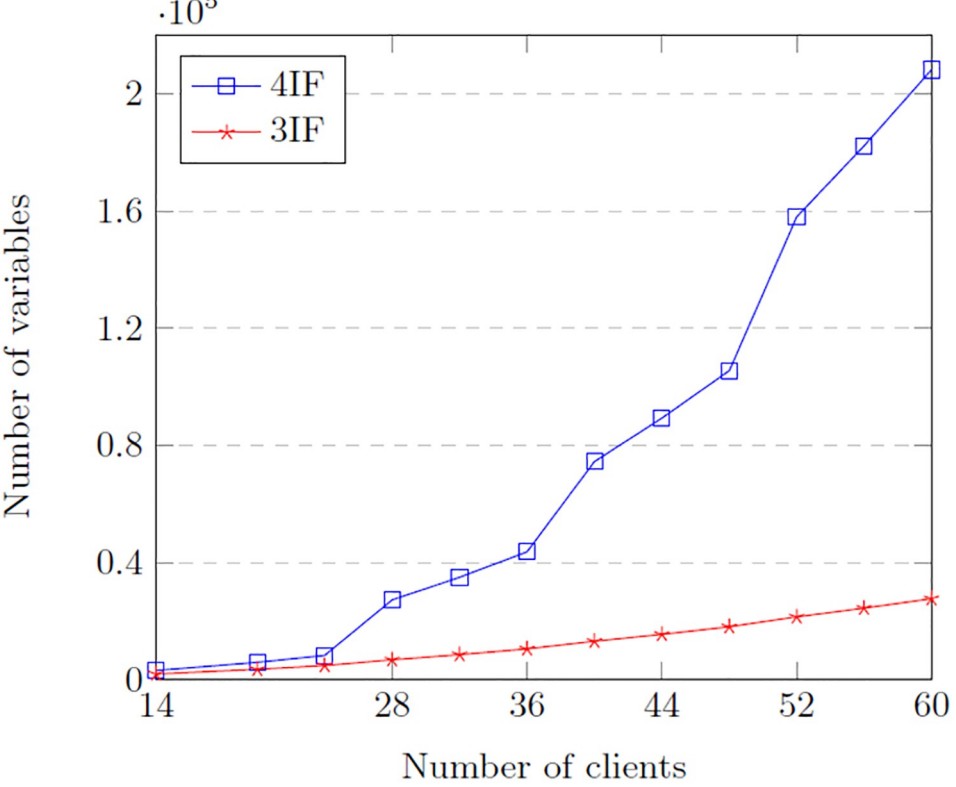

**Fig 2. Integer variables in both formulations.**

MDHFPCVRP-TW. All the experiments were run on a computer with an Intel Core i7 4 GHz processor with 64 GB RAM and Ubuntu as the operating system. The modeling was done in Python and solved using GUROBI. All the solver's default parameters were used.

This section is divided into two parts: initially, performance over generated data sets are discussed. Then, the performance of both formulations over a real-life case study is discussed.

## Generated data sets

To evaluate the performance of both formulations we decided to create our own data set, given that there are no data sets reported in the literature with the conditions we required. Also, the real-life case study is very large and we suspected that not many conclusions could be reached from that data set alone. All the information required to generate the data sets can be found in https://github.com/aarenas2/MDHFPCVRP-TW.

We decided to generate the data sets randomly locating the clients in a $100 \times 50$ grid and using the euclidean distance for the parameter $t$. Two depots are always located in the center of the left and right side of the grid (in a similar fashion to the real-life case study). To compare the performance of the 4IF and 3IF we designed various data sets varying the number of total visits required ($\Xi$) and the clients' frequencies of visit. These last factor could be set as: dense (the majority of the clients must be visited more than three times per week), sparse (the majority of the clients must be visited less than three times per week), and balanced. The purpose to evaluate this last factor was to find out if the total number of clients (and of nodes) was more critical than the total number of visits (which is directly associated with the total number of vehicles), or if the contrary was correct. The clients' demand was generated with a discreet uniform distribution between two and eight vending machines per client which resembles the real-life case study.

Other parameters were not considered as factors because they are fixed or proportional to $\Xi$. Specifically:

- Depots' capacity: it was set to $\dfrac{2\Xi}{3}$ in all the data sets.

- Number of vehicles: it was set to $\dfrac{\Xi}{10}$ in all the data sets.

- Time windows length: it was set to 120 units of time in all the data sets.

- Admitted stand-by time: it was set to 30 units of time in all the data sets.

- Maximum duration of the route: it was set to 300 units of time in all the data sets.

To compare the performance and scalability of the formulations, we solved data sets with every combination of clients' frequencies of visit for $\Xi$ equal to 30, 60, 90, and 120. To avoid obtaining biased results, we solved every possible combination five times with different clients' distribution on the grid, and different demands. The time limit was set to 3600 seconds. In each case, we recorded the relative GAP percentage (when the formulation found optimal solutions, the number in parenthesis indicates how many were found), the required time to find the first feasible solution (TFFFS) in seconds, and the number of times no feasible solution was found (NFS). Let us present the obtained results in Tables 5 to 8:

With the presented results, it is easy to see that both formulations scale badly due to the number of integer variables and constraints with integer variables that they must process. The 3IF was capable of finding some feasible solutions in the largest data sets. On the other hand, the 4IF could only find one feasible solution out of 15 in the same type of data sets. It can be also seen that both formulations are more sensible to the number of clients than to the number

**Table 5. Results with Ξ = 30.**

| Parameters | | Avg. GAP (%) | | Avg. TFFFS (s) | | NFS | |
|---|---|---|---|---|---|---|---|
| Distribution | Clients | 4IF | 3IF | 4IF | 3IF | 4IF | 3IF |
| Dense | 8 | 0.6% [4] | 0.0% [5] | 1.0 | 1.0 | 0/5 | 0/5 |
| Balanced | 11 | 2.0% [2] | 1.1% [3] | 11.4 | 1.6 | 0/5 | 0/5 |
| Sparse | 13 | 3.2% | 1.5% [3] | 24.8 | 4.0 | 0/5 | 0/5 |

**Table 6. Results with Ξ = 60.**

| Parameters | | Avg. GAP (%) | | Avg. TFFFS (s) | | NFS | |
|---|---|---|---|---|---|---|---|
| Distribution | Clients | 4IF | 3IF | 4IF | 3IF | 4IF | 3IF |
| Dense | 16 | 3.8% | 4.8% | 61.0 | 30.2 | 0/5 | 0/5 |
| Balanced | 22 | 6.1% | 5.1% | 73.4 | 79.8 | 0/5 | 0/5 |
| Sparse | 26 | 9.5% | 12.3% | 486.0 | 164.2 | 0/5 | 0/5 |

**Table 7. Results with Ξ = 90.**

| Parameters | | Avg. GAP (%) | | Avg. TFFFS (s) | | NFS | |
|---|---|---|---|---|---|---|---|
| Distribution | Clients | 4IF | 3IF | 4IF | 3IF | 4IF | 3IF |
| Dense | 24 | 6.4% | 9.3% | 430.6 | 152.6 | 0/5 | 0/5 |
| Balanced | 33 | 10.0% | 7.7% [a] | 2805.0 | 453.0 [a] | 4/5 | 0/5 |
| Sparse | 39 | 17.4% | 11.5% [a] | 1605.5 | 612.5 [a] | 3/5 | 0/5 |

[a] computed using the data sets were the 4IF found a feasible solution.

**Table 8. Results with Ξ = 120.**

| Parameters | | Avg. GAP (%) | | Avg. TFFFS (s) | | NFS | |
|---|---|---|---|---|---|---|---|
| Distribution | Clients | 4IF | 3IF | 4IF | 3IF | 4IF | 3IF |
| Dense | 32 | 7.8% | 7.2% [a] | 1766.0 | 1048.0 [a] | 4/5 | 0/5 |
| Balanced | 44 | - | 15.0% | - | 1774.7 | 5/5 | 2/5 |
| Sparse | 52 | - | 14.8% | - | 78.5 | 5/5 | 3/5 |

[a] computed using the data sets were the 4IF found a feasible solution.

of visits (Ξ) and vehicles. The 4IF found a feasible solution in all the data sets with 24 clients and 90 visits, but for the same number of visits and more clients, the formulation struggled to find feasible solutions. Something similar happened to the 3IF with 120 visits were it could always find a feasible solution with 32 clients but struggled with more.

Deciding which formulation bare better relative GAPs is not straightforward. This is due to the mixed results found in the experimentation. On the other hand, the 3IF is clearly better suited to find quicker a feasible solution as it outperformed the 4IF most of the times in this indicator on this experimentation. If we combine this with the fact that the 3IF found a feasible solution in 92% of the data sets versus the 4IF's 65%, it leaves the 3IF as the better of the two formulations according to the results of this experimentation.

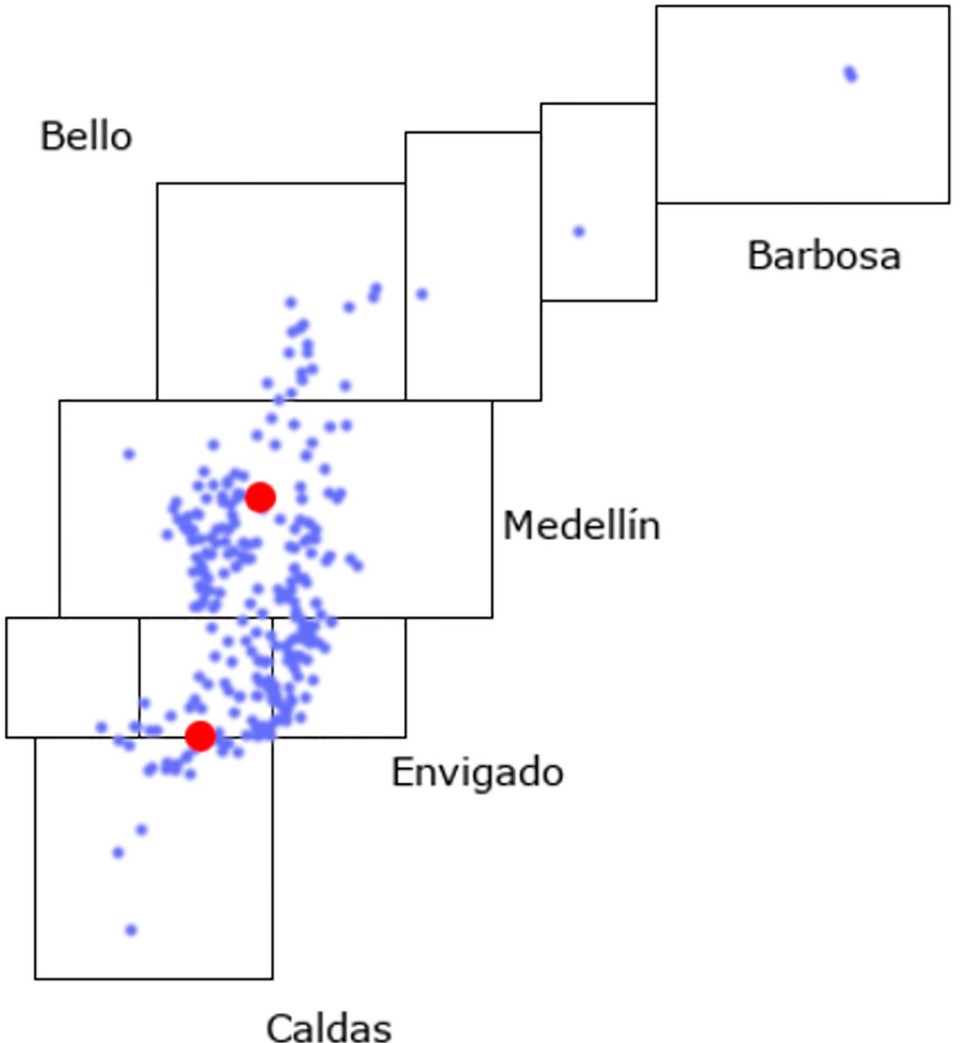

**Fig 3. Nodes' location in real-life data set.**

## Real-life data set

The vending machine company in Medellín, Colombia, has to solve every week a MDHFPCVRP-TW with two depots (one in the south and one in the north of the metropolitan area of Medellín), 262 clients with demands ranging from two to 10 vending machines each. 104(39.7%) clients must be visited daily, 87(33.2%) clients thrice per week, 49(18.7%) twice per week, and the rest (8.4%) only once. 88(33.6%) clients have no time windows, 47 (17.9%) have 120 minutes long time windows, and 127(48.5%) have 240 minutes long time windows. The maximum allowed stand-by time varies from client to client but is either zero minutes, half an hour, or an hour. The operation must be performed in an eight hour span. The average times to travel between nodes were calculated using a paid mapping system with the geo coordinates of each node. The company has 22 vehicles with the capacity to serve 12 vending machines and 45 vehicles with the capacity to serve 16 vending machines. The data set called *MedellinVending262* is stored in https://github.com/aarenas2/MDHFPCVRP-TW. In Fig 3 the location of the clients (blue dots) and depots (red dots) are presented.

Both formulations were run for 12 hours. Both exhausted the local memory of the CPU were all the experimentation were conducted. The 4IF did so in the 483rd second during the presolve step. The 3IF did so in the 2240th second when it was solving the root node.

## Conclusions

In this article, we presented two different but equivalent formulations for a PVRP with time windows, multiple depots, and heterogeneous fleet, not equal to any seen with these same attributes, considering all the particularities associated with a real-life case study in the vending machine industry in Medellín, Colombia. Also, the high incidence of integer variables growth is shown in both formulations, when the number of customers and available vehicles increases.

We performed a series of experiments to comprehend if the two equivalent formulations bear different performances in GUROBI. Based on our experimentation, the results let us conclude that using 3IF or 4IF affects the end result in the solver. For small data sets, both achieve similar relative GAP but the 3IF finds quicker an initial feasible solution. When the size of the data set is augmented, the problem escalates badly, but the 3IF finds feasible solutions easier than the 4IF. As a consequence, using 3IF is more reliable as more clients must be attended. Neither of the formulations can be used in the real-life case study due to its size so a different approach must be considered.

The two formulations presented in this research can be used to solve VRP problems with periodicity, time windows, heterogeneous fleet, or multiple depots easily without having to include all of them. For instance, removing multiple depots requires $|D|$ to be one. Eliminating periodicity requires $|H|$ to be one. To avoid considering time windows, the problem requires assigning to every client long enough times to be visited. Removing heterogeneous fleet requires all vehicles to have the same capacity.

We also presented an analysis of papers with PVRP that included the attributes of our formulation. The optimization models of this kind analyzed in this paper were compared, and their characteristics were summarized. Even though all of the five problems analyzed include the same attributes (in some cases more), they are unique in their way. Our problem is closer to those presented in [9, 12] as they share 16 of the 23 individual characteristics. But they are far from equal. In fact, our problem is the only one that allows stand-by times before a time window and has a maximum time for a route.

In future work, we will add valid inequalities to the formulations to have better results in larger data sets. We also want to develop a branch-and-cut algorithm to find a tighter GAP using exact methods. Furthermore, we want to comprehend the effects of the formulations when the number of clients is fixed and only the frequencies of visits are altered. Lastly, we want to know how our formulations perform in problems with some of the four studied attributes.

## Supporting information

**S1 Appendix.**
(PDF)

## Author Contributions

**Conceptualization:** Alejandro Arenas-Vasco, Juan Carlos Rivera, Maria Gulnara Baldoquín.

**Data curation:** Alejandro Arenas-Vasco.

**Formal analysis:** Alejandro Arenas-Vasco, Juan Carlos Rivera, Maria Gulnara Baldoquín.

**Investigation:** Alejandro Arenas-Vasco, Juan Carlos Rivera.

**Methodology:** Alejandro Arenas-Vasco, Juan Carlos Rivera.

**Project administration:** Juan Carlos Rivera.

**Resources:** Alejandro Arenas-Vasco.

**Software:** Alejandro Arenas-Vasco, Juan Carlos Rivera.

**Supervision:** Juan Carlos Rivera, Maria Gulnara Baldoquín.

**Validation:** Maria Gulnara Baldoquín.

**Visualization:** Alejandro Arenas-Vasco.

**Writing – original draft:** Alejandro Arenas-Vasco.

**Writing – review & editing:** Alejandro Arenas-Vasco, Juan Carlos Rivera, Maria Gulnara Baldoquín.

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
