## [Decision Letter · Decision Letter 0]

23 May 2024

PONE-D-24-18029Effect of formulations over a Periodic Capacitated Vehicle Routing Problem with Multiple Depots, Heterogeneous Fleet, and Hard Time-WindowsPLOS ONE

Dear Dr. Arenas-Vasco,

Thank you for submitting your manuscript to PLOS ONE. After careful consideration, we feel that it has merit but does not fully meet PLOS ONE’s publication criteria as it currently stands. Therefore, we invite you to submit a revised version of the manuscript that addresses the points raised during the review process.

The paper is well-written. The authors should clarify how they produce the instances and ensure their accessibility. In addition, additional experimental evaluation is necessary to reinforce the main contribution. All reviewers'comments should be carefully considered and solved in your revised manuscript.

We look forward to receiving your revised manuscript.

Kind regards,

Yangming Zhou, Ph.D.

Academic Editor

PLOS ONE

Journal Requirements:

2. Please note that PLOS ONE has specific guidelines on code sharing for submissions in which author-generated code underpins the findings in the manuscript. In these cases, we expect all author-generated code to be made available without restrictions upon publication of the work. Please review our guidelines at https://journals.plos.org/plosone/s/materials-and-software-sharing#loc-sharing-code and ensure that your code is shared in a way that follows best practice and facilitates reproducibility and reuse."

"The authors want to thank Universidad EAFIT who has financed the research project

“Un estudio poliedral y m´etodos de soluci´on para problemas de enrutamiento de veh´ıculos multi-atributo con m´ultiples periodos en entornos urbanos” with internal

code 953-000020, in which this study has been developed. The authors also want to

thank Colfuturo who financed partially this project with the grant “Convocatoria"

Please remove any funding-related text from the manuscript and let us know how you would like to update your Funding Statement. Currently, your Funding Statement reads as follows: "The authors want to thank Universidad EAFIT (www.eafit.edu.co) who has financed the research project "Un estudio poliedral y métodos de solución para problemas de enrutamiento de vehículos multi-atributo con múltiples periodos en entornos urbanos'' with internal code 953-000020, in which this study has been developed. The authors also want to thank Colfuturo (www.colfuturo.org) who financed partially this project with the grant "Convocatoria 909''. All grants were given to Arenas-Vasco A.

Sponsors did not play any role in the study design, data collection and analysis, decision to publish, or preparation of the manuscript."Please include your amended statements within your cover letter; we will change the online submission form on your behalf.

Additional Editor Comments :

This paper presents two new problem formulations for the Periodic Vehicle Routing Problem (PVRP). The paper is well-written. I have the following comments and suggestions which may further improve the manuscript. The authors should clarify how they produce the instances and ensure their accessibility. In addition, additional experimental evaluation is necessary to reinforce the main contribution.

Reviewers' comments:

Reviewer's Responses to Questions

**Comments to the Author**

1. Is the manuscript technically sound, and do the data support the conclusions?

Reviewer #1: No

Reviewer #2: Yes

2. Has the statistical analysis been performed appropriately and rigorously? 

Reviewer #1: No

Reviewer #2: Yes

3. Have the authors made all data underlying the findings in their manuscript fully available?

Reviewer #1: Yes

Reviewer #2: Yes

4. Is the manuscript presented in an intelligible fashion and written in standard English?

Reviewer #1: Yes

Reviewer #2: Yes

5. Review Comments to the Author

Reviewer #1: The authors introduced two different but equivalent formulations for a PVRP with time windows, multiple depots, and heterogeneous fleet, considering all the particularities associated with a real-life scenario in the vending machine industry in Medellin, Colombia. The authors also performed some experiments to comprehend if the two equivalent formulations bear different performances in GUROBI.

1.The studied topic is practical. The paper is well-written. But the main contribution of this work is to propose two formulations based on classical VRP models for a real-life problem. The workload of the paper seems not enough. I suggest this work to be published in another journal.

Reviewer #2: The authors present a novel contribution to the Periodic Vehicle Routing Problem (PVRP) by introducing two new problem formulations that differ in the usage of the crucial flow variable. Experiments have demonstrated their impact on the performance of the GUROBI solver. The topic is interesting, and the manuscript is well-structured. There is only a small issue: the authors should explain in detail how they produce the instance data and ensure its accessibility. Additionally, more information is needed regarding the results on the real cases mentioned in the paper.

6. PLOS authors have the option to publish the peer review history of their article (what does this mean?). If published, this will include your full peer review and any attached files.

Reviewer #1: No

Reviewer #2: No

---

## [Author Response · Author response to Decision Letter 0]

22 Jul 2024

Medellín, 5th July of 2024

Academic Editor: Yangming Zhou, Ph.D.

PLOS ONE

First of all, we want to thank you, and the reviewers for all the valuable comments received about the manuscript of the paper "Effect of formulations over a Periodic Capacitated Vehicle Routing Problem with Multiple Depots, Heterogeneous Fleet, and Hard Time-Windows." We found them very useful and we changed the structure of the paper to comply with the observations. 

We have prepared an improved paper version, considering the reviewers' comments, which we will explicitly reply to next. All the changes are written in blue. The section “Performance comparison between formulations” was completely changed.

General comments: 

Comment (C)/ “The paper is well-written. The authors should clarify how they produce the instances and ensure their accessibility. In addition, additional experimental evaluation is necessary to reinforce the main contribution. All reviewers'comments should be carefully considered and solved in your revised manuscript.”

Response (R)/ Section “Performance comparison between formulations” now clarifies how the instances were produced. Also, a more complete experimentation was done and analyzed. In addition, a real-life case study data set was added.

C/ “Journal requirements.”

R/ All the journal requirements are now met (PLOS ONE's style, code sharing guidelines, funding is not in acknowledgments anymore, instances are available).

C/ 1. Is the manuscript technically sound, and do the data support the conclusions?

Reviewer #1: No

Reviewer #2: Yes

R/ A more complete experimentation was performed and included in section “Performance comparison between formulations.”

C/ 2. Has the statistical analysis been performed appropriately and rigorously?

Reviewer #1: No

Reviewer #2: Yes

R/ In the section “Performance comparison between formulations,” the different possible factors that might influence the results were taken into account.

C/ 3. Have the authors made all data underlying the findings in their manuscript fully available?

Reviewer #1: Yes

Reviewer #2: Yes

R/ Additional to tables in the paper, all the instances were made available on GitHub.

C/ 4. Is the manuscript presented in an intelligible fashion and written in standard English?

Reviewer #1: Yes

Reviewer #2: Yes

R/ All the changes made to the original manuscript followed the same writing style as the original manuscript.

C/ Reviewer #1: The authors introduced two different but equivalent formulations for a PVRP with time windows, multiple depots, and heterogeneous fleet, considering all the particularities associated with a real-life scenario in the vending machine industry in Medellin, Colombia. The authors also performed some experiments to comprehend if the two equivalent formulations bear different performances in GUROBI.

1.The studied topic is practical. The paper is well-written. But the main contribution of this work is to propose two formulations based on classical VRP models for a real-life problem. The workload of the paper seems not enough. I suggest this work to be published in another journal.

R/ Additional to the new formulations of the model, the proposed problem is new and conclusions over the performance of both formulations are presented.

We also added a more complete experimentation to further strengthen our conclusions. We also presented the real-life case study data set. 

C/ Reviewer #2: The authors present a novel contribution to the Periodic Vehicle Routing Problem (PVRP) by introducing two new problem formulations that differ in the usage of the crucial flow variable. Experiments have demonstrated their impact on the performance of the GUROBI solver. The topic is interesting, and the manuscript is well-structured. There is only a small issue: the authors should explain in detail how they produce the instance data and ensure its accessibility. Additionally, more information is needed regarding the results on the real cases mentioned in the paper.

R/ The process to generate the instances is now detailed and accessible via GitHub. Additionally, information about the real case instances is provided in the second subsection of “Performance comparison between formulations.”

C/ While revising your submission, please upload your figure files to the Preflight Analysis and Conversion Engine (PACE) digital diagnostic tool, https://pacev2.apexcovantage.com/. PACE helps ensure that figures meet PLOS requirements. To use PACE, you must first register as a user. Registration is free. Then, login and navigate to the UPLOAD tab, where you will find detailed instructions on how to use the tool. If you encounter any issues or have any questions when using PACE, please email PLOS at figures@plos.org. Please note that Supporting Information files do not need this step.

R/ Images not generated using latex (Fig 1 and 3) were uploaded after being adapted in PACE.

Best Regards,

Articles’s authors.

---

## [Decision Letter · Decision Letter 1]

9 Sep 2024

Effect of formulations over a Periodic Capacitated Vehicle Routing Problem with Multiple Depots, Heterogeneous Fleet, and Hard Time-Windows

PONE-D-24-18029R1

Dear Dr. Arenas-Vasco,

We’re pleased to inform you that your manuscript has been judged scientifically suitable for publication and will be formally accepted for publication once it meets all outstanding technical requirements.

Kind regards,

Prof. Yangming Zhou,

Additional Editor Comments (optional):

The authors well address all issues

Reviewers' comments:

Reviewer's Responses to Questions

**Comments to the Author**

1. If the authors have adequately addressed your comments raised in a previous round of review and you feel that this manuscript is now acceptable for publication, you may indicate that here to bypass the “Comments to the Author” section, enter your conflict of interest statement in the “Confidential to Editor” section, and submit your "Accept" recommendation.

Reviewer #1: All comments have been addressed

Reviewer #2: All comments have been addressed

2. Is the manuscript technically sound, and do the data support the conclusions?

Reviewer #1: Yes

Reviewer #2: (No Response)

3. Has the statistical analysis been performed appropriately and rigorously? 

Reviewer #1: Yes

Reviewer #2: (No Response)

4. Have the authors made all data underlying the findings in their manuscript fully available?

Reviewer #1: Yes

Reviewer #2: (No Response)

5. Is the manuscript presented in an intelligible fashion and written in standard English?

Reviewer #1: Yes

Reviewer #2: (No Response)

6. Review Comments to the Author

Reviewer #1: The authors indicated that the proposed problem is new and provided a more complete experimentation to further strengthen the conclusions. In addition, a real-life case study data set is provided. I think that the authors have addressed all my concerns, and recommend the paper to be published.

Reviewer #2: (No Response)

7. PLOS authors have the option to publish the peer review history of their article (what does this mean?). If published, this will include your full peer review and any attached files.

Reviewer #1: No

Reviewer #2: No

---

## [Editor Report · Acceptance letter]

18 Sep 2024

PONE-D-24-18029R1 

PLOS ONE

Dear Dr. Arenas-Vasco, 

I'm pleased to inform you that your manuscript has been deemed suitable for publication in PLOS ONE. Congratulations! Your manuscript is now being handed over to our production team.

Kind regards, 

on behalf of

Prof. Yangming Zhou 

Academic Editor

PLOS ONE